# Evaluating the Accuracy of Upper Limb Movement in the Sagittal Plane among Computer Users during the COVID-19 Pandemic

**DOI:** 10.3390/healthcare12030384

**Published:** 2024-02-02

**Authors:** Arkadiusz Jaskólski, Ewa Lucka, Mateusz Lucki, Przemysław Lisiński

**Affiliations:** Department of Rehabilitation and Physiotherapy, University of Medical Sciences, 28 Czerwca 1956 Str., No 135/147, 60-545 Poznań, Poland; jaskolskiarek@ump.edu.pl (A.J.); mat539@interia.pl (M.L.); plisinski@vp.pl (P.L.)

**Keywords:** upper limb movement, sagittal plane, computer users, COVID-19 pandemic

## Abstract

(1) Background: The most common musculoskeletal pathology among healthcare professionals is neck and/or shoulder pain. The aim of this study was to determine the dominant upper limb functionality concerning the ability to replicate a given movement pattern among employees reporting neck or upper limb pain while using a computer during the COVID-19 pandemic. (2) Methods: The study was conducted from March to April 2021 on a group of 45 medical employees who used a computer workstation for 4 to 6 h of their working time. In the design of this study, three study groups were created: a group of patients with pain syndrome of segment C5/C7 of the spine, a group of patients with shoulder pain syndrome, and a control group of healthy volunteers. (3) Results: The examined groups significantly differed in the correctness of performing the given movement (*p* = 0.001) and the minimum value of inclination during the exercise session (*p* = 0.026), as well as the maximum lowering (*p* = 0.03) in relation to the control group. (4) Conclusions: The VECTIS device can be used to assess the accuracy of reflecting the prescribed movement of the upper limb in rehabilitation programs for patients with cervical spine pain syndrome and shoulder pain syndrome.

## 1. Introduction

The COVID-19 pandemic has had a major impact on the way we work, with many people now working from home or in other remote locations [1]. As a result, there has been an increased need for medical workers to use computers for their work, which can lead to increased risk of upper limb musculoskeletal disorders (ULMDs) [2]. In the period preceding the COVID-19 pandemic, as many as 40–80% of employees who used a computer continuously in the course of their duties had musculoskeletal dysfunctions [3,4,5,6]. During the pandemic, among healthcare professionals using computers at work, the most common pathology of the musculoskeletal system was pain in the cervical spine and/or shoulder associated with incorrect ergonomics of the workplace and working time (long working hours, prolonged use of the computer, sedentary work, lack of changes in posture during work) [7,8]. At the same time, the duration of pain depended to a large extent on the age of employees and seniority [9]. Neck and shoulder pain was a recurring disorder as 60–80% of employees experienced another episode a year after the initial episode [6]. It is worth noting that the assessment of the patient’s ability to replicate a given upper limb movement, which was the subject of the study, pertained to hospital workers who, during the pandemic, often continuously provided medical services despite the ailments and discomfort or pain resulting from working on the computer. The research direction presented in this paper aligns with employer expectations. From the employer’s perspective, in order to prevent the aforementioned phenomena, the implementation of preventive and rehabilitative measures at the early stage of musculoskeletal overload was crucial [10,11]. Neck and/or shoulder pain negatively affected the limbs used by an employee to operate a computer, both in terms of static and dynamic work [12,13]. Therefore, an important element of diagnostic activities is the objective assessment of the function of the aforementioned limb as a “specific link” between the employee and the computer, especially during arm movements similar to those performed when using a computer [14].

It is worth noting that pain reported in the shoulder girdle or cervical spine has a subjective nature. Relatively little is known about the relationships between the use of the dominant upper limb and pain reported by computer users. In other words, we do not know how the dominant upper limb functions in individuals who report pain related to more intense and longer computer use. In the available literature, there are relatively few research results describing this phenomenon. Most studies focused on assessing the range of motion of the upper limb using a goniometer [1,15] and motion sensors [16,17]. The evaluation of the correctness of upper limb movements has so far been conducted using electroencephalographic (EEG) signals for brain–computer interface in patients with spinal cord injuries [18]. The innovative approach to assessing the kinesthetic and proprioceptive function of the upper limb involved using a robotic arm for dynamic movement reproduction. However, the addition of visual information (i.e., observing arm movements) had varied effects on task performance [19]. While not discounting the validity of the methods described above in the study of upper limb functionality, it seems essential to create a diagnostic study algorithm with a higher degree of accuracy and enabling active patient participation in therapy using this diagnostic device.

The aim of the study was to determine the dominant upper limb functionality concerning the ability to replicate a given movement pattern among employees reporting neck or upper limb pain while using a computer for remote work during the COVID-19 pandemic.

## 2. Materials and Methods

### 2.1. Study Design

The subjects’ computer usage during the pandemic extended to 4 to 6 h per day. Three research groups were created for this study: a group of patients with cervical pain syndrome (SpG), a group of patients with shoulder pain syndrome (ShG), and a control group of healthy patients (CG). In the first stage of the study, employees who reported the onset of neck or shoulder pain during the pandemic and had not been previously diagnosed were included. In the next stage, all employees meeting the criteria underwent a screening clinical examination conducted by a medical rehabilitation doctor and an orthopedist to confirm the association with the onset of symptoms during the pandemic and to diagnose diseases related to computer usage at work. The clinical examination included physical and diagnostic assessments: magnetic resonance imaging of the cervical spine and shoulder. Based on the clinical examination results, employees were classified into one of three groups: the first group consisted of individuals with no diagnosed pain—the control group (CG)—the second group consisted of individuals diagnosed with cervical spine pain syndrome—the cervical group (SpG)—and the third group consisted of employees with clinically confirmed painful shoulder syndrome (ShG). In the next stage of the study, all participants were assessed on the VECTIS measuring device to evaluate the precision of reproducing the given movement using the same research protocol. Each group first conducted a trial test. The next day, the test was repeated, and the results are presented in the article. The flowchart of the study project is shown in Figure 1.

### 2.2. Participants

The control group (CG) consisted of 15 employees (9 females and 6 males), who, during the physical and clinical examination, did not exhibit symptoms of cervical or shoulder pain syndrome. The inclusion criteria for the study were an extension of computer work hours from 4 to 6 h during the COVID-19 pandemic, the absence of neck and shoulder pain complaints, as well as other pain complaints reported before and during the pandemic, and good overall health confirmed by clinical examination. Exclusion criteria included employees who, before the pandemic, performed computer-related duties on a full-time basis, those whose computer work hours during the pandemic did not involve an extension, and employees who, in the interview, reported a history of musculoskeletal injury or neurological disorders.

The study group with cervical pain syndrome (SpG) comprised 15 patients (13 females, 2 males), in whom clinical and imaging examinations, based on MRI, diagnosed cervical pain syndrome, indicating disc herniation at the C5/C7 level. The inclusion criteria for the study were an extension of computer work hours from 4 to 6 h during the pandemic, the onset of neck pain during the pandemic, confirmation of the association between neck pain and prolonged computer work by physical examination, and muscle strength of neck and shoulder muscles above 4. Exclusion criteria included a chronic history of cervical pain syndrome, patients who had undergone cervical spine surgery, patients with carpal tunnel syndrome, and patients who had experienced a stroke. Also excluded were patients with pre-pandemic neck pain confirmed in imaging studies conducted before the pandemic (discopathies, surgeries of the cervical and lumbar spine, carpal tunnel syndrome, previous stroke). Additionally, patients were excluded if physical examination revealed abnormalities in the natural curvature of the spine (flattening of lordosis or kyphosis of the cervical spine, increased kyphosis of the thoracic spine, flattening or kyphosis of the lumbar spine), patients with diagnosed rheumatic diseases (RA, AS), compression fractures of the vertebrae, and employees with muscle strength of neck and shoulder muscles below 4 on the Lovett scale.

The study group with shoulder pain syndrome (ShG) comprised 15 patients (11 females, 4 males) clinically diagnosed with shoulder pain syndrome confirmed by ultrasound/MRI of the rotator cuff muscles. Inclusion criteria for the study were an extension of computer work hours from 4 to 6 h associated with the pandemic, the presence of shoulder girdle pain during or after work that was not present before the pandemic, and a Lovett scale score > 4, indicating that muscle strength allows the patient to perform movement against resistance greater than the weight of their own limb. Exclusion criteria included a chronic nature of shoulder girdle pain before the pandemic, previous shoulder injuries and fractures of the humerus, limb paralysis due to a stroke, and a Lovett scale score < 4, indicating that the patient could not perform movement against resistance provided only by the weight of the limb.

### 2.3. Measuring Instrument

The study used the device VECTIS SN 0112/2016, manufactured by AC International EAST, intended to rehabilitate the upper limb. The device is used to assess the accuracy of replicating a given movement pattern. Systems operating on a similar principle aim to precisely measure the accuracy of movement and provide the user with feedback, such as a score or graphical representation of movement accuracy [20]. Additionally, the device can be programmed to provide additional feedback to the user, such as the direction of movement, movement speed, or the force applied to the lever. These systems find application in physiotherapy and rehabilitation, allowing users to easily and accurately monitor their progress. By providing feedback and encouraging the user to adjust their movements, the system can help improve accuracy and efficiency [21]. Systems assessing the ability to replicate a designated movement path are also used as incentives for users to continue exercising, as they can track their progress and strive to improve their performance each time they use the device.

The VECTIS device operates on the principle of feedback using flexible resistance elements. A small resistance was generated in the initial phase of the movement, which was increased evenly in the later phase of the exercise. The test sessions lasted two minutes and included a program aimed at the accuracy of the given movement. By lifting the lever of the device, the subject was supposed to reflect the path of movement displayed on the device’s monitor, which resembled a sinusoidal ribbon. To perform the given movement, the subject performed a smooth movement consisting of a combination of alternating flexion in the shoulder joint (Figure 2) and extension in the shoulder joint (Figure 3). 

### 2.4. Data Collection

The analysis of the characteristics of the study groups included the registration of sociodemographic variables such as gender, age, body weight, height, and BMI.

For the analysis of parameters assessing the performance of the assigned movement, the following were the data recorded on the measuring device parameters: accuracy of mapping the set movements (movement path visible on the monitor); ranges of movements: the value of the minimum and maximum deflection, the average and maximum change during lifting (corresponding to flexion in the shoulder joint), the average and maximum time of lifting, the value of the upper average and maximum change during rest, upper value of average and maximum time during rest, average and maximum value of change during lowering, average and maximum lowering time (corresponding to shoulder extension), lower values of average and maximum time during rest, and lower values of average and maximum change during rest. Figure 4 depicts the schematic representation of the significance of individual parameters.

### 2.5. Statistical Analysis

Data were analyzed using Statistica version 13.1 (StatSoft Co., Krakow, Poland). Descriptive statistics were given as means and standard deviations (SDs), and categorical variables were given as counts. The Shapiro–Wilk test was used to assess the normality of distributions. Non-parametric analyses were used when the data did not meet the assumptions of the parametric analysis. To assess the significance of differences between the study results (the group with spinal pain syndrome C and the group with painful shoulder syndrome) and the control group, the parametric t-Student test, the Welch test (with a lack of homogeneity of variance), or the non-parametric Mann–Whitney test were used. Post hoc analysis was used where there were statistically significant differences in the measurements. The chi-square test was used to compare differences between groups for categorical variables. *p*-values less than 0.05 were considered statistically significant.

To calculate the sample size, a statistical formula was applied, assuming a significance level (α) of 0.05, a test power (1-β) of 95%, and a variance of 50%. The minimum required sample size for each group was 15 individuals.

### 2.6. Research Ethics

The study was conducted in accordance with the ethical principles of biomedical research as defined in the Declaration of Helsinki. Each participant was informed of the purpose and methodology of the study and gave informed consent to participate. The study was prospective and was approved by the Ethics Committee of the Medical University of Karola Marcinkowski in Poznań (Consent No. 320/22 of 14 April 2022).

## 3. Results

The study groups significantly differed in BMI (*p* = 0.01) and its components: body weight (*p* < 0.0001) and height (*p* = 0.018). The average age of patients with cervical pain syndrome was 58.93 ± 12.94 years, while those with shoulder pain syndrome were 63.2 ± 13.65 years old, and the healthy controls were 64.8 ± 3.74 years old. In addition, patients from the control group presented a normal weight according to BMI, while patients with cervical pain syndrome and shoulder pain syndrome presented as overweight. A detailed summary of the group characteristics is presented in Table 1.

The studied groups differed significantly in the accuracy of mapping the given motion (*p* = 0.001). The accuracy of mapping the movement path displayed on the device’s monitor showed that the subjects from the control group improved by 83.8 ± 14.81% of the set motion. Meanwhile, patients with pain syndrome segment C performed only 59 ± 18.73% correctly, while patients with painful shoulder syndrome performed correctly 57.4 ± 17.59% of the set motion. The examined groups also differed in the minimum value of the inclination during the exercise session (*p* = 0.026): the subjects from the control group reached a value of 16.33 ± 9.08 mm, while the patients with pain syndrome C-segment achieved 31.4 ± 14.43 mm, and patients with painful shoulder syndrome reached a value of 32.07 ± 20.66 mm. Patients with spinal segment C pain syndrome had significantly longer times of maximum lifting (*p* = 0.02) and maximum lowering (*p* = 0.03) in relation to the control group. Detailed results of the parameters measured by the VECTIS device are presented in Table 2.

## 4. Discussion

A full understanding of movement quality in a joint requires the integration of kinematic (movement) and kinetic (force) analyses to identify internal forces (e.g., from muscles, ligaments) and external forces (e.g., resulting from the load acting on the joint) [22,23]. The aim of our study was to assess the ability to replicate a designated movement in medical professionals whose computer work hours were extended due to the COVID-19 pandemic. Over the years, various methods have been used to analyze the nature of the movement in the shoulder joint, evaluating various aspects of the biomechanics of the joint: range, speed, power, and work [24,25,26]. Most of these methods were used for research purposes, such as the Biodex device, which evaluates the biomechanics of movement in isokinetic conditions or muscle strength in isometric conditions [27]. The VECTIS device used in our study to measure the accuracy of the mapping of the given motion uses the mechanism of resistance of elastic bands during the exercise session. It showed that patients with cervical spine pain syndrome and shoulder pain syndrome differed in the parameters of reproducing the assigned movement compared to healthy individuals. The tested groups differed significantly in the correctness of performing the given movement (*p* = 0.001) and the minimum value of inclination during the exercise session (*p* = 0.026) (Table 2). It is worth noting that VECTIS works based on visual feedback. Gueye et al. [28] proved that using a rehabilitation program with visual biofeedback is effective in assessing the correctness of performing a given task during an exercise session in relation to conventional methods. In turn, Breen et al. [29] have proven that using a biofeedback system can effectively correct neck posture in computer users. In the qualitative method of assessing shoulder movement that we used, the measurement of its range plays an important role. Moreira et al. [30] and Gosain et al. [31] demonstrated that non-ergonomic work at a computer station may cause pain in the upper limb due to the increased shoulder abduction angle. With regard to the occurrence of neck pain, the time of performing a specific motor task is also important. It is worth noting that the obtained results of the reproduced movement, measured in the sagittal plane, reflect the position of the closed kinematic chain during computer work. Although computer users may appear to only use their arm, due to the fact that the movement occurs in conditions of a closed kinematic chain, the muscles of the shoulder girdle play a significant role in stabilizing the scapula and upper limb to ensure maximum functionality of the upper limb. The results obtained on the device in a closed kinematic system have significant implications in clinical practice regarding the design of computer workstations to take into account the proper kinematics of the upper limb during computer work. Kotani et al. [32] found that arm rotation ranges up to 25 degrees of internal rotation when using a keyboard and up to 15 degrees of external rotation when using a mouse. During the exercise, employees diagnosed with pain syndrome of the cervical spine obtained significantly longer times of maximum lifting (i.e., extension in the shoulder joint; *p* = 0.02) and maximum lowering (i.e., flexion in the shoulder joint; *p* = 0.03) in relation to people without pain (Table 2). Although in the study we analyzed the accuracy of upper limb movement reconstruction separately for people with neck pain and separately for shoulder pain, in clinical practice these two locations of pain often coexist. It has been proven that neck pain caused by changing the tension of the muscles of the neck and shoulder girdle disturbs coordination and affects the quality of movement in the shoulder. Lin et al. [33] proved that the occurrence of cervical spine pain syndrome in office workers affects the cervical section’s biomechanics and the upper limb’s range of motion. Similarly, Amiri et al. [34] proved that the diagnosis of limited range of motion and deep sensation in chronic cervical pain may have a predictive value in preventing muscle weakness in both the cervical spine and the shoulder complex muscles.

In conclusion, the limitations of our study stem from the inclusion of a relatively small sample size of 45 medical employees, potentially restricting the generalizability of the results. Additionally, the short study duration underscores the need for long-term observations to offer a more comprehensive understanding. Furthermore, the absence of comparative data from before the pandemic hampers the contextualization of our findings. Due to the methodological and technical capabilities of the research instrument used, the analysis focused on movement in the sagittal plane. However, the conclusions drawn from the results encourage consideration of undertaking a similar evaluation in other anatomical planes.

In summary, the study demonstrates several strengths that contribute to its significance in the field of rehabilitation. Despite the limited number of participants (45 healthcare workers), this group represents a cross-section of computer users, providing valuable insights into the impact of remote work on upper limb mobility. It is worth emphasizing that the study’s findings highlight the utility of assessing the accuracy of shoulder joint movement in the diagnosis and rehabilitation of patients with dysfunction in the cervical spine and shoulder girdle. A positive aspect of the study is the identification of movement accuracy disorders in individuals currently experiencing neck and shoulder pain, which has practical implications for therapists and patients. Furthermore, although references to the scientific literature are based on articles from several years ago, they provide context and significance to the study, adding scientific value. The significant differences observed in the accuracy of movement between the studied groups further confirm the importance of the research outcomes. In conclusion, these strengths underscore the value of the study in expanding knowledge about the impact of remote work on upper limb mobility and highlight its practical potential in the field of rehabilitation.

## 5. Conclusions

Using the VECTIS device, a rehabilitation program can be designed, and the range of motion in the shoulder joint of patients with cervical spine pain syndrome and shoulder pain syndrome can be assessed. Patients with cervical spine pain syndrome and shoulder pain syndrome exhibited differences in the range of motion parameters in the shoulder joint, as evaluated by VECTIS, compared to healthy individuals. The assessment of the precision of execution in relation to the accuracy of reproducing the assigned movement pattern was a differentiating factor between individuals with neck and shoulder pain and those without pain symptoms. This implies the consideration of the above fact in the development of improvement algorithms for patients with cervical spine and shoulder pain syndromes.

## Figures and Tables

**Figure 1 healthcare-12-00384-f001:**
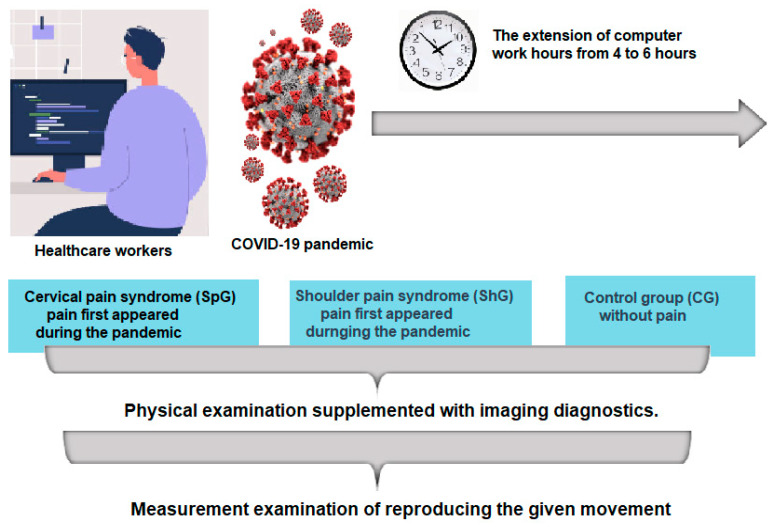
Study protocol.

**Figure 2 healthcare-12-00384-f002:**
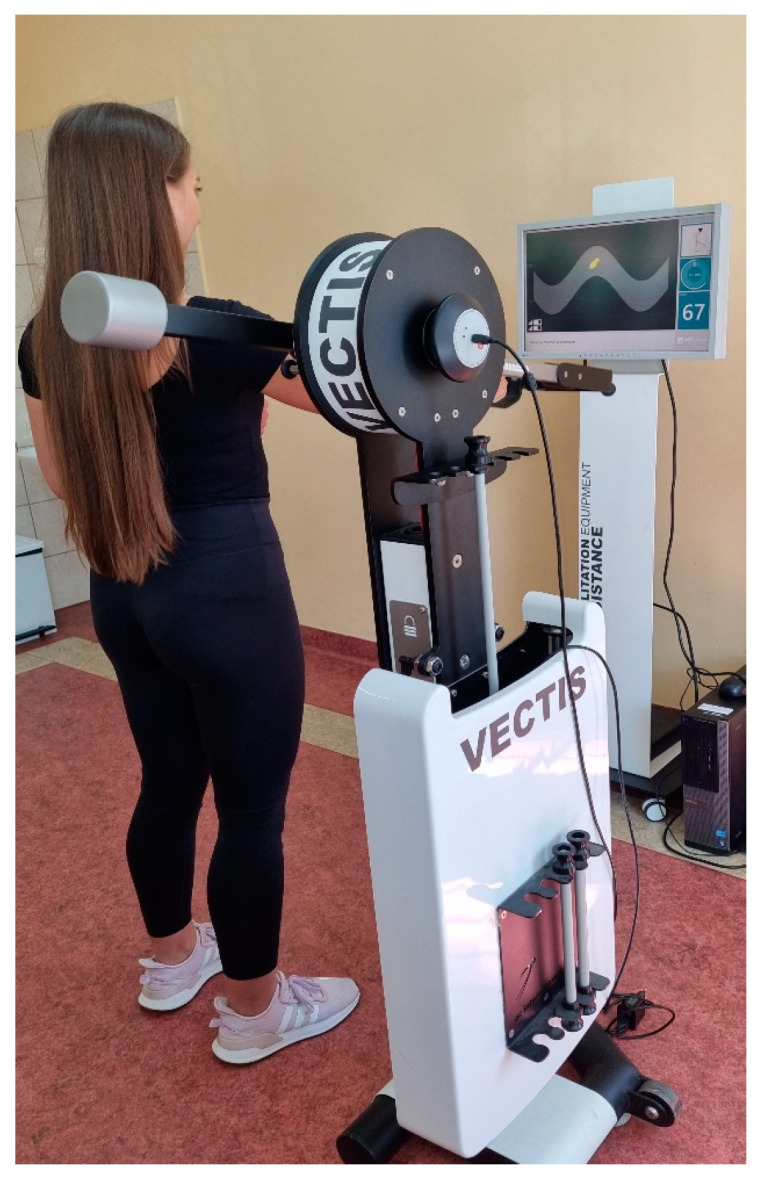
Flexion during lifting.

**Figure 3 healthcare-12-00384-f003:**
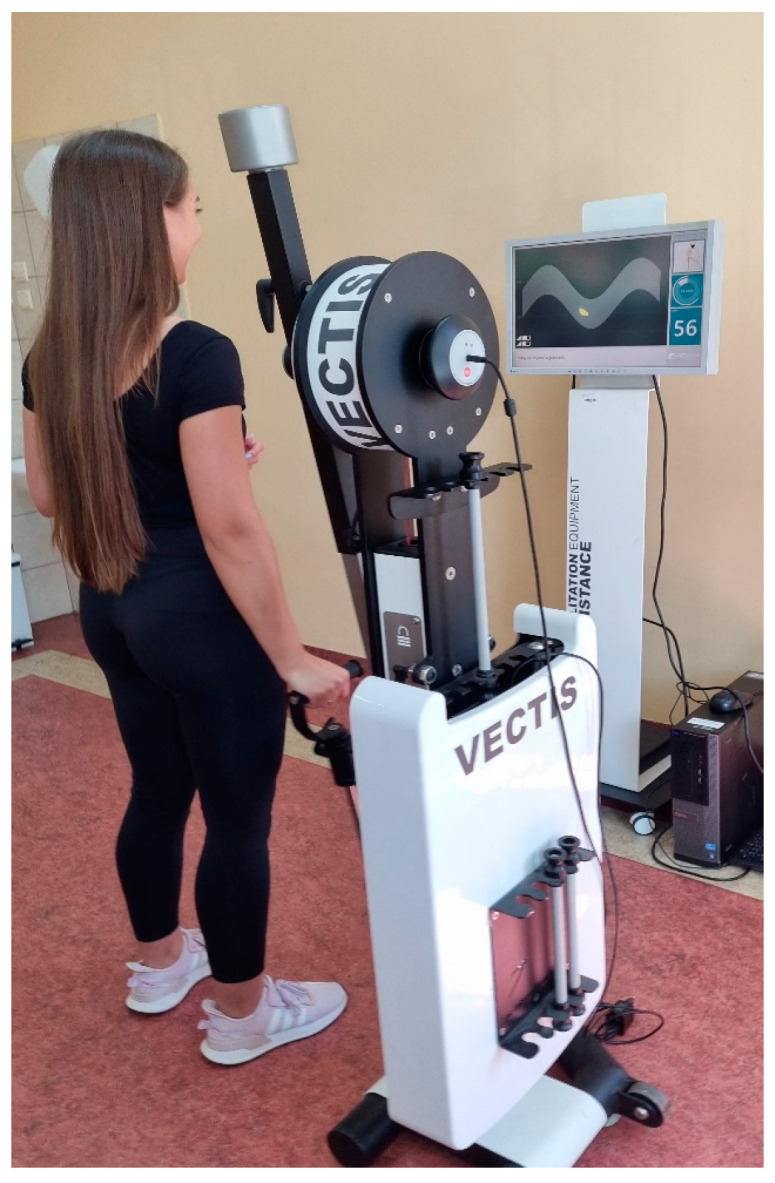
Extension during lowering.

**Figure 4 healthcare-12-00384-f004:**
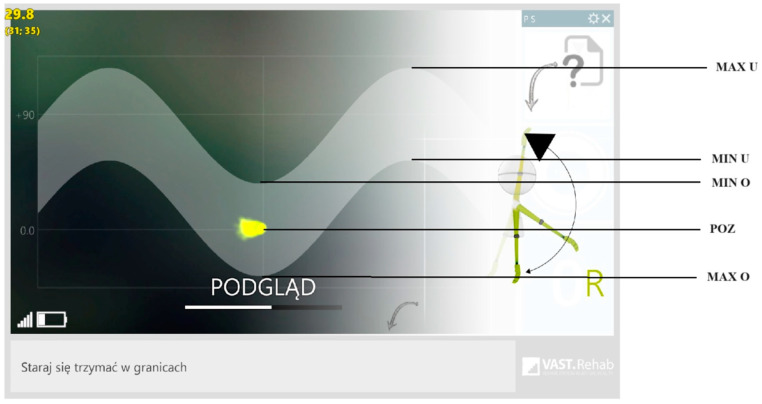
View of the task path displayed on the device. MAX U—Maximum flexion value (lifting) in the shoulder joint. MIN U—Minimum flexion value (lifting) in the shoulder joint. MIN O—Minimum extension value (lowering). POZ—Position of the upper limb under examination. MAX O—Maximum extension value (lowering).

**Table 1 healthcare-12-00384-t001:** Demographic and clinical data.

		Study Groups			*p*
		CG	SpG	ShG	
Gender n (%)	Females	9 (60.00%)	13 (86.67%)	11 (73.33%)	0.26 ^a^
	Males	6 (40.00%)	2 (13.33%)	4 (26.67%)	
Age (years)	Mean ± SD	64.8 ± 3.74	58.93 ± 12.94	63.2 ± 13.65	0.93 ^b^
	Median	66	64	67	
	Min–max	58–70	27–76	24–78	
Body weight (kg)	Mean ± SD	72.27 ± 11.58	72.2 ± 10.84	76.3333 ± 11.62	<0.0001 ^b^
	Median	74	72	74	
	Min-max	55–94	54–96	59–98	
Height (cm)	Mean ± SD	1.73 ± 0.09	1.66 ± 0.07	1.66 ± 0.07	0.018 ^b^
	Median	1.72	1.68	1.66	
	Min–max	1.62–1.95	1.56–1.79	1.56–1.78	
BMI	Mean ± SD	24.04 ± 2.56	26.199 ± 3.44	27.68 ± 3.21	0.01 ^b^
	Median	23.77	27.22	28.23	
	Min–max	20.15–27.68	18.69–31.23	19.71–33.91	

^a^ Chi^2^ test; ^b^ Mann–Whitney test BMI-Body mass index CG—control group, SpG—spinal syndrome group, ShG—shoulder syndrome group, n—size of the sample; *p*—probability value, SD—standard deviation.

**Table 2 healthcare-12-00384-t002:** Analysis of parameter values measured during the training session on the VECTIS device in the study groups.

		Study Groups			*p* ^b^			
		CG	SpG	ShG	*p*	SpG vs. CG	ShG vs. CG	SpG vs. ShG
Accuracy of mapping the given motion (%)	Mean ± SD	83.8 ± 14.81	59 ± 18.73	57.4 ± 17.59	0.001	0.003	0.002	1
Minimum deflection value (mm)	Mean ± SD	16.33 ± 9.08	31.4 ± 14.43	32.07 ± 20.66	0.026	0.04	0.09	1
Maximum deflection value (mm)	Mean ± SD	108 ± 11.84	109.4 ± 13.71	108.93 ± 18.31	0.9	1	1	1
Value of the average change during lifting (mm)	Mean ± SD	87.2 ± 7.73	94.18 ± 12.23	91.45 ± 15.59	0.3	0.28	0.61	0.82
Value of the maximum change during lifting (mm)	Mean ± SD	115.06 ± 13.29	131.51 ± 23.23	128.1 ± 27.79	0.11	0.12	0.26	0.91
Average lifting time (s)	Mean ± SD	1.25 ± 0.11	1.34 ± 0.1	1.32 ± 0.13	0.09	0.09	0.24	0.86
Maximum lifting time (s)	Mean ± SD	2.19 ± 0.84	2.54 ± 0.5	2.26 ± 0.43	0.026	0.02	0.43	0.65
Value of the upper average change during rest (mm)	Mean ± SD	1.5 ± 2.38	2.79 ± 3.11	1.39 ± 2.08	0.27	0.78	1	0.51
Value of the upper maximum change during rest (mm)	Mean ± SD	1.87 ± 3.04	3.06 ± 3.36	2.07 ± 3.31	0.39	0.83	1	0.83
Value of the upper average time during rest (s)	Mean ± SD	0.02 ± 0.03	0.04 ± 0.05	0.02 ± 0.03	0.16	0.36	1	0.49
Value of the upper maximum rest time (s)	Mean ± SD	0.02 ± 0.04	0.05 ± 0.05	0.03 ± 0.05	0.26	0.49	1	0.78
Value of the average change during leaving (mm)	Mean ± SD	91.03 ± 5.7	99.1 ± 13.28	96.84 ± 13.48	0.17	0.18	1	0.97
Value of the maximum change during descent (mm)	Mean ± SD	112.1 ± 11.87	130.72 ± 23.88	129.48 ± 27.52	0.06	0.08	0.23	1
Average leaving time (s)	Mean ± SD	1.55 ± 0.07	1.54 ± 0.1	1.47 ± 0.15	0.28	1	0.43	0.59
Maximum leave time (s)	Mean ± SD	2.23 ± 0.59	2.97 ± 1	2.64 ± 0.99	0.04	0.03	0.47	0.79
Value of the lower average change during rest (mm)	Mean ± SD	0.9 ± 0.44	0.58 ± 0.77	0.68 ± 0.81	0.17	0.26	0.49	1
Value of the lower maximum change during rest (mm)	Mean ± SD	1.2 ± 0.86	0.87 ± 1.27	1.29 ± 2.07	0.25	0.36	0.71	1
Value of the lower average time during rest (s)	Mean ± SD	0.01 ± 0.01	0.01 ± 0.01	0.01 ± 0.01	0.17	0.33	0.41	1
Value of the lower maximum rest time (s)	Mean ± SD	0.02 ± 0.01	0.01 ± 0.02	0.02 ± 0.03	0.24	0.38	0.64	1
Energy expenditure during the session (kcal.)	Mean ± SD	2.73 ± 0.19	2.82 ± 0.46	2.79 ± 0.62	0.48	1	1	0.68

^b^ Mann–Whitney test; BMI—body mass index CG—control group, SpG—spinal syndrome group, ShG—shoulder syndrome group, n—size of the sample; *p*—probability value, SD—standard deviation.

## Data Availability

The data presented in this study are available on request from the first author. The data are not publicly available due to ethical restrictions.

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
