# Peer review of "Evaluating the Accuracy of Upper Limb Movement in the Sagittal Plane among Computer Users during the COVID-19 Pandemic"

_healthcare, 2024, doi:10.3390/healthcare12030384_

Round 1
Reviewer 1 Report
Comments and Suggestions for Authors
Dear authors,
I hope this letter finds you well. I have carefully reviewed your manuscript titled "Evaluating the Accuracy of Upper Limb Movement in the Sagittal Plane Among Computer Users during the COVID-19 Pandemic” and I would like to provide you with my feedback and suggestions for improvement. Overall, I find the topic of your study to be relevant and interesting. However, there are several issues that I would like to address:
Introduction:
Lines 53-55 and 64-66: A bibliographic reference supporting this statement is needed.
There is no clear justification supporting the study's objective. The authors should establish a framework that aids in understanding what is known about this topic and why conducting the present study is necessary.
Moreover, I understand that lines 68-71 correspond to the study's objective. However, there has been no prior discussion of the VECTIS device, resistance-elastic bands, or exercise. It is necessary to clarify the study's objective. Additionally, the title indicates that the study took place during the COVID-19 Pandemic, so this information should be included in the objective.
Materials and methods:
Subsection "2.1. Study Design and Participants" y "2.2. Participants". Why do both subsections refer to participants? Unify the information related to participants in a single subsection.
2.1. Study Design and Participants:
What is the study design?
Lines 74-91: This information should not appear in this subsection. Moreover, a bibliographic reference supporting this paragraph is needed.
2.2. Participants:
How was the sample size calculated for each group?
This section should only address participants, inclusion criteria, groups, etc. Data collection should go in another subsection titled "Data Collection," for example.
The number of women and men included in each group, as well as their average age, are sociodemographic data that should appear in the results section, not in this section.
A reference for the Lovett scale is needed.
Results:
In the materials and methods, it should be included that sociodemographic variables such as gender, age, body weight, height, and BMI were recorded.
What are the primary variables?
Additionally, the material and methods section should define the different variables measured by the VECTIS device for a better understanding by the reader.
Discussion:
The first paragraph of the discussion should begin by recalling the study's objective and presenting the main findings. It is intriguing that the authors dedicate a paragraph to discussing a descriptive variable of the sample, such as BMI. Analyzing this variable did not require the use of the VECTIS device, so I understand that identifying differences between groups in BMI was not one of the study's objectives.
What implications do the results of the present study have for clinical practice?
What are the strengths and limitations of the study? This should be the last paragraph of your discussion.
Overall, the discussion is very weak, probably because there are many weaknesses throughout the study.
Conclusion: “The Vectis device can be used to assess the accuracy of reflecting the prescribed movement of the upper limb…” This is not a validation study of the VECTIS device, so this conclusion is not supported by the study's results. Furthermore, it is challenging to establish a conclusion when there is no clear description of the study's objectives. I recommend restructuring this section first.
I appreciate the effort you have put into this research, and I believe that addressing these issues will significantly improve the quality of your manuscript.
Best regards.
Author Response
Thank you very much for your time and valuable suggestions regarding the correction of the article. We tried to make corrections according to the guidelines, and we hope that the article is now more clear and refined
Points1: Introduction: Lines 53-55 and 64-66: A bibliographic reference supporting this statement is needed.
There is no clear justification supporting the study's objective. The authors should establish a framework that aids in understanding what is known about this topic and why conducting the present study is necessary.
Moreover, I understand that lines 68-71 correspond to the study's objective. However, there has been no prior discussion of the VECTIS device, resistance-elastic bands, or exercise. It is necessary to clarify the study's objective. Additionally, the title indicates that the study took place during the COVID-19 Pandemic, so this information should be included in the objective.
Response 1: Thank you for the valuable feedback. We have made revisions to the introduction and added a paragraph outlining the study's objective. We believe that the introduction section is now more transparent.
Ponts 2 Materials and methods:
Subsection "2.1. Study Design and Participants" y "2.2. Participants". Why do both subsections refer to participants? Unify the information related to participants in a single subsection.
2.1. Study Design and Participants:
What is the study design?
Lines 74-91: This information should not appear in this subsection. Moreover, a bibliographic reference supporting this paragraph is needed.
2.2. Participants:
How was the sample size calculated for each group?
This section should only address participants, inclusion criteria, groups, etc. Data collection should go in another subsection titled "Data Collection," for example.
The number of women and men included in each group, as well as their average age, are sociodemographic data that should appear in the results section, not in this section.
A reference for the Lovett scale is needed.
Response 2: Thank you very much for the valuable feedback. We have made modifications to the section, changed the names of subsections, and made corrections to ensure clarity of the data. We added a description of the research project and included a study diagram for data transparency. Additionally, we added a bibliographic reference to the device description. In the statistical analysis section, we explain the selection of the sample size, expanded the description of the research groups with clear explanations of inclusion and exclusion criteria, and included information regarding the Lovett scale.
Point 3 :Results:
In the materials and methods, it should be included that sociodemographic variables such as gender, age, body weight, height, and BMI were recorded.
What are the primary variables?
Additionally, the material and methods section should define the different variables measured by the VECTIS device for a better understanding by the reader.
Response 3 Thank you for the valuable feedback. We have made corrections as suggested. In the methodology section, we added a subsection on data collection, including information on the registration of sociodemographic variables such as gender, age, body mass, height, and BMI. We also added a diagram illustrating the significance of individual parameters measured during the execution of movements.
Point 4 Discussion:
The first paragraph of the discussion should begin by recalling the study's objective and presenting the main findings. It is intriguing that the authors dedicate a paragraph to discussing a descriptive variable of the sample, such as BMI. Analyzing this variable did not require the use of the VECTIS device, so I understand that identifying differences between groups in BMI was not one of the study's objectives.
What implications do the results of the present study have for clinical practice?
What are the strengths and limitations of the study? This should be the last paragraph of your discussion.
Overall, the discussion is very weak, probably because there are many weaknesses throughout the study.
Response 4 Thank you for the suggestions. We removed the sentence about BMI, added an explanation regarding the clinical implications of the obtained results, and included a paragraph outlining the strengths and weaknesses of the study
Point 5 Conclusion: “The Vectis device can be used to assess the accuracy of reflecting the prescribed movement of the upper limb…” This is not a validation study of the VECTIS device, so this conclusion is not supported by the study's results. Furthermore, it is challenging to establish a conclusion when there is no clear description of the study's objectives. I recommend restructuring this section first.
Response 5 We made corrections to the conclusions section
Reviewer 2 Report
Comments and Suggestions for Authors
Hello,
I believe your has 2 different themes. On one hand, according to the title, you are studying the "accuracy of upper limb movement in the sagittal plane among computer users during the COVID-19 pandemic." However, on the other hand, in some areas of your paper the reader is led to believe the goal of the paper is to somehow validate the use of the Vectis device for measuring the prescribed motion. Please be sure to clearly focus on the goal and desired aim of the paper.
In regards to patient screening, the average age of your patients is the early/mid 60's. Did you account for any kyphotic changes in their posture which may impact their ability to lift their arm in the sagittal plane? Did you account for how long each person was in their respective position, and if they needed to use a computer for long periods prior to Covid?
* Line 61- remove "implemented" or "used"
* Lines 110-114 and 115-119 are duplicate
* lines 122-123- did all 15 people in the C group have C5/7 disc herniations? Any other conditions?
* line 121 you state the avg age of the c-spine pain group as 58.93 however line 161 you state 64 years; shoulder in line 129 is listed as 63 years but in 161 it is 67 years; controls 64.8 years in line 116 but 66 years in line 162. Suggest listing true averages or change listing from "average age...." to "median age...." in this section
Thank you for your time and best of luck!
Comments on the Quality of English LanguageMinor corrections as stated above
Author Response
Thank you very much for your time and valuable suggestions regarding the correction of the article. We tried to make corrections according to the guidelines, and we hope that the article is now more clear and refined
I believe your has 2 different themes. On one hand, according to the title, you are studying the "accuracy of upper limb movement in the sagittal plane among computer users during the COVID-19 pandemic." However, on the other hand, in some areas of your paper the reader is led to believe the goal of the paper is to somehow validate the use of the Vectis device for measuring the prescribed motion. Please be sure to clearly focus on the goal and desired aim of the paper.
In regards to patient screening, the average age of your patients is the early/mid 60's. Did you account for any kyphotic changes in their posture which may impact their ability to lift their arm in the sagittal plane? Did you account for how long each person was in their respective position, and if they needed to use a computer for long periods prior to Covid?
Response 1: Thank you for your suggestions. We have made modifications in the methodology and materials section, expanded the description of the research group
* Line 61- remove "implemented" or "used"
* Lines 110-114 and 115-119 are duplicate
* lines 122-123- did all 15 people in the C group have C5/7 disc herniations? Any other conditions?
* line 121 you state the avg age of the c-spine pain group as 58.93 however line 161 you state 64 years; shoulder in line 129 is listed as 63 years but in 161 it is 67 years; controls 64.8 years in line 116 but 66 years in line 162. Suggest listing true averages or change listing from "average age...." to "median age...." in this section
Response 2: Thank you for your attention, we made corrections,
Reviewer 3 Report
Comments and Suggestions for Authors
1. Please kindly arrange citations chronologically in the text, following the Instructions for Authors. https://www.mdpi.com/journal/healthcare/instructions
2. So far, there has been no study evaluating the correctness of performing a given movement using the VECTIS device, taking into account the additional use of resistance-elastic bands during an exercise session. Please rephrase this sentence and turn it into the novel elements of the study.
3. At the end of the introduction, please include the purpose of the study and specify it in the abstract.
4. The control group consisted of 15 employees 110 (9 women and 6 men) with an average age of 64.8 ± 3.74 years is repeated twice in the text. Please kindly revise.
5. Please specify the exclusion criteria for the control group as well.
6. In the first sentence of the discussion, please relate to the study's objectives. Which they were and whether they were achieved.
Comments on the Quality of English LanguageModerate editing of the English language is needed.
Author Response
Thank you very much for your time and valuable suggestions regarding the correction of the article. We tried to make corrections according to the guidelines, and we hope that the article is now more clear and refined
Point 1: Please kindly arrange citations chronologically in the text, following the Instructions for Authors. https://www.mdpi.com/journal/healthcare/instructions
Response 1:
Thank you for your suggestions. We have applied corrections to the citations according to the MDPI author guidelines, including citation numbering in text.
Point 2: So far, there has been no study evaluating the correctness of performing a given movement using the VECTIS device, taking into account the additional use of resistance-elastic bands during an exercise session. Please rephrase this sentence and turn it into the novel elements of the study.
Response 2 :
Thank you for the suggestions. We have made corrections to the introduction.
Point 3. At the end of the introduction, please include the purpose of the study and specify it in the abstract.
Response 3
Thank you for the suggestions. We have added the research aim in the introduction and abstract.
Point 4 The control group consisted of 15 employees 110 (9 women and 6 men) with an average age of 64.8 ± 3.74 years is repeated twice in the text. Please kindly revise.
Response 4 We removed the sentence, thank you for the suggestions.
Point 5 Please specify the exclusion criteria for the control group as well.
Response 5 : Thank you for the suggestions; we expanded the methodology section, providing a detailed description of the study group, including inclusion and exclusion criteria
Point 6 In the first sentence of the discussion, please relate to the study's objectives. Which they were and whether they were achieved.
Response 6 Thank you for the suggestions. We added a reference to the study objectives
Point 7. Comments on the Quality of English Language Moderate editing of the English language is needed.
Response 7 We made corrections to the English language.
Round 2
Reviewer 1 Report
Comments and Suggestions for Authors
Dear authors,
Thank you for the changes made to your manuscript. My final recommendation would be to remove the sentence mentioning the objective in lines 79-82, as it duplicates information with lines 106-109.
Best regards.
Author Response
Dear reviuver
Thank you for the suggestion. I have incorporated the correction in line with your advice.
I wanted to express my sincere gratitude for your invaluable feedback and dedicated time spent reviewing my work. Your insightful corrections have significantly enriched the content and quality of the manuscript. Once again, thank you for your invaluable support and for being an integral part of this collaborative effort. Your expertise and dedication are immensely appreciated.
Warm regards
Reviewer 2 Report
Comments and Suggestions for Authors
Very well done.
The adjustments you have provided have made significant improvements.
Best of luck!
Author Response
Dear Reviewer, I wanted to express my sincere gratitude for your invaluable feedback and dedicated time spent reviewing my work. Your insightful corrections have significantly enriched the content and quality of the manuscript. Once again, thank you for your invaluable support and for being an integral part of this collaborative effort. Your expertise and dedication are immensely appreciated. Warm regards